# Evolutionary Aspects of TRPMLs and TPCs

**DOI:** 10.3390/ijms21114181

**Published:** 2020-06-11

**Authors:** Dawid Jaślan, Julia Böck, Einar Krogsaeter, Christian Grimm

**Affiliations:** Walther Straub Institute of Pharmacology and Toxicology, Faculty of Medicine, Ludwig-Maximilians-Universität, 80336 Munich, Germany; Dawid.Jaslan@lrz.uni-muenchen.de (D.J.); julia.boeck@lrz.uni-muenchen.de (J.B.); einar.krogsaeter@cup.unimuenchen.de (E.K.)

**Keywords:** endosome, lysosome, TPC1, TPC2, TRPML1, TRPML2, TRPML3, TRP

## Abstract

Transient receptor potential (TRP) or transient receptor potential channels are a highly diverse family of mostly non-selective cation channels. In the mammalian genome, 28 members can be identified, most of them being expressed predominantly in the plasma membrane with the exception of the mucolipins or TRPMLs which are expressed in the endo-lysosomal system. In mammalian organisms, TRPMLs have been associated with a number of critical endo-lysosomal functions such as autophagy, endo-lysosomal fusion/fission and trafficking, lysosomal exocytosis, pH regulation, or lysosomal motility and positioning. The related non-selective two-pore cation channels (TPCs), likewise expressed in endosomes and lysosomes, have also been found to be associated with endo-lysosomal trafficking, autophagy, pH regulation, or lysosomal exocytosis, raising the question why these two channel families have evolved independently. We followed TRP/TRPML channels and TPCs through evolution and describe here in which species TRP/TRPMLs and/or TPCs are found, which functions they have in different species, and how this compares to the functions of mammalian orthologs.

## 1. Introduction

The earliest evidence for life on Earth dates back some 3.5 billion years [1]. First, prokaryotes (bacteria and archaea) and certain eukaryotes (protozoa, algae, and fungi) evolved, then gradually plants and animals started developing (their evolution started about 5–25 million years ago). The diversity and differential ability to adapt to distinct environments enabled different species to eventually inhabit even the most extreme areas of planet Earth [2]. The development of a range of sensory modalities was particularly essential for them to navigate through an ever-changing environment [2]. Transient receptor potential (TRP) channels, which mediate cation flow down their electrochemical gradients in response to various environmental stimuli, play fundamental roles for perception in most living organisms. As “cellular sensors”, they are critical for, e.g., vision, olfaction, taste, and temperature-, mechano-, and osmosensation [3,4,5].

The founding member of the TRP family was initially discovered as a receptor-operated sensory cation channel in a blind strain of *Drosophila melanogaster* [6,7,8], which paved the way for the discovery of the canonical (TRPC1–TRPC7) subfamily of mammalian TRPs [9,10]. Beyond the TRPC channels, the other 21 members of the mammalian TRP family can be grouped in five branches, based on sequence homology: vanilloid (TRPV1-TRPV6), melastatin (TRPM1-8), ankyrin (TRPA1), mucolipin (TRPML1-3), and polycystin (TRPP1-3). Additionally, the TRP channels can be split into two major groups: Group I (TRPC, TRPV, TRPM, and TRPA), and Group II (TRPML and TRPP). The Group II channels appear highly homologous in their transmembrane domains, and express a unique long loop between transmembrane segments S1 and S2 [11]. The existence of subfamilies which are not found in mammals, TRPN (NOMPC-like) expressed in some vertebrates and TRPVL (VL = voltage-like; Cnidaria and *Capitella teleta*), indicates that TRP channels have been somewhat plastic during metazoan evolution [12].

A common feature of all TRPs is their homotetrameric assembly of six-transmembrane subunits or domains (S1–S6). Unlike other described ion channel families, activation and regulatory mechanisms of TRP channels cannot be reliably predicted based on their subfamily memberships. Often TRPs are activated by multiple stimuli, suggesting that the physiologically relevant stimulus for any given TRP will be governed by the specific cellular context [13,14,15,16,17,18]. Therefore, a functional classification of TRPs may seem more informative. An alternative classification of TRP channels proposed by Zang et al. (2018) combines its physiological function with endogenous activation mechanisms, leading to three subgroups: metabotropic, sensory, and organellar TRPs. Thereby, a functional subgroup can contain members from different subfamilies, important when comparing TRP channels from different species. Fundamental physiological and cellular functions of TRP channels may be the reason for them being highly conserved in yeast and mammals. Nevertheless, TRP channel genes seem to be completely absent from land-plant genomes [12]. In this review, we discuss evolutionary aspects of organellar TRP channels, in particular the TRPML/MCOLN channels, and the functionally related two-pore channels (TPCs).

## 2. TRPMLs—More and More over Time

Accidental discoveries are often the most significant ones. Such a fortunate accident occurred in 1955, when Christian de Duve discovered the lysosome (Greek: “digestive body”) while attempting to localize the enzyme glucose-6-phosphatase [19]. The lysosome is traditionally understood as an intracellular macromolecule degradation center with a very low pH (approximately 4.5), maintained by lysosomal membrane proton pumps (v-ATPase). Furthermore, the lysosomal digestive capacity is maintained by more than 60 types of acidic hydrolases residing in its lumen [20]. Traditionally, lysosomes are considered to maintain cellular health upon removing toxic cellular components, recycling damaged organelles, terminating signal transduction cascades, and maintaining metabolic homeostasis. Recent studies have added to this view, highlighting their fundamental role as signaling hubs and as hosts of the major nutrient sensors in the cell [21].

Regulated ion fluxes across the lysosomal membrane are a crucial driving force for a number of processes, providing the ionic environment necessary for nutrient homeostasis, osmotic adaptation, or lysosomal trafficking and fusion [22]. Several ion channels controlling the flux of Cl^−^, K^+^, Na^+^, and Ca^2+^ ions are embedded within lysosomal membranes. Activation of, e.g., the TRPMLs facilitates lysosomal cation efflux, decreasing the lysosomal membrane potential and facilitating v-ATPase proton pumping. Resultantly, the lysosomal resting membrane potential lies between −40 and −20 mV [23,24,25]. These estimates are however a contentious topic, with positive membrane potentials also having been reported [26].

The endo-lysosomal channels have recently gained increasing attention as drug targets implicated in various pathologies [23,27,28,29,30,31,32,33,34]. TRPMLs which are highly critical for endosomal/lysosomal function and autophagy [35,36,37,38] form a separate branch within the superfamily of TRP channels. The TRPML channel family is comprised of three members in mammalian genomes, which share about 75% amino acid sequence similarity [38]. TRPML1, TRPML2, and TRPML3 expression can be detected in early, late, and recycling endosomes as well as in lysosomes to various degrees, allowing flux of cations such as Ca^2+^ into the cytosol from the endo-lysosomal lumen [11]. All three TRPMLs are activated by the phosphoinositide PI(3,5)P_2_, a major constituent of endo-lysosomal membranes, and by the luminal pH/proton concentration which differs between various endosomal organelles and lysosomes [27,32,39]. TRPML1 is expressed and present in all tissues, while expression of TRPML2 and TRPML3 appears cell-type specific, indicative of tissue-specific functions of these isoforms [40,41]. Genetic mutations resulting in loss of TRPML1 function by channel inactivation or mislocalization cause a rare genetic disorder called Mucolipidosis Type IV (MLIV), after which the channel initially was named (mucolipin-1, MCOLN1). Loss of MCOLN1 channel function results in a detrimental lysosomal storage disease marked by infantile mental retardation, corneal opacities, strabismus, and delayed motor development [42]. On the other hand, murine gain-of-function TRPML3 mutants result in the varitint-waddler phenotype, marked by hearing and pigmentation defects [43,44,45,46]. Diseases related to mutations in TRPML2 have not yet been described, although the TRPML2 knock-out mouse shows impaired macrophage mobilization attributing to impaired chemoattractant (CCL2) release [32,47].

The number of TRPML channels expressed varies between different species. Thus, studying their evolution may help to better understand their physiological roles and functions. The TRPML1 gene (*MCOLN1*) is highly conserved in the animal kingdom. Furthermore, many species have more than one TRPML gene. All known genomes of tetrapods have three copies of TRPML channels, with the exception of the domesticated pig (*Sus scrofa*) where *Mcoln3* was found to be a severely truncated pseudogene, lacking five of the six transmembrane domains and the pore-coding region [48]. The human *MCOLN1* gene is localized on chromosome 19, distinct from *MCOLN2* and *MCOLN3*, which are closely located on chromosome 1 [49,50]. The *MCOLN2* gene is located downstream in the same orientation as *MCOLN3*, in the genomes of most jawed vertebrates (Gnathostomes), but not pigs, *Xenopus tropicalis*, and zebrafish [50].

Some deuterostomes as well as protostomes (e.g., *Strongylocentrotus purpuratus*, *Branchiostoma floridae*, *Ciona intestinalis*, *Capitella teleta*, and *Drosophila melanogaster*) contain one *Mcoln* gene ortholog each (Figure 1). García Añoveros et al. (2014) described three episodes of *Mcoln* gene multiplications in the Nephrozoa clade. In 2017, the inshore hagfish (*Eptatretus burgeri*) genome was sequenced [51], revealing that the jawless fish genome encodes only one *Mcoln* homolog (Figure 1—phylogenetic tree; orange branches). It resultantly becomes evident that duplications of the ancestral *Mcoln* gene, resulting in three paralogs, occurred between the onset of craniates and jawed vertebrates (Gnathostomes) (Figure 1—phylogenetic tree; orange and green branches).

It is interesting to note that homology and localization of the *Mcoln2* and *Mcoln3* genes suggest their origin resulting from unequal crossing-over in a common ancestor, placing both gene copies in tandem [50]. Similar evolutionary events occurred in the case of closely linked *Tmem138* and *Tmem216* in tetrapod genomes but not in *Xenopus tropicalis* and zebrafish [52]. The most recent evolutionary *Mcoln* multiplication took place in zebrafish (*Danio rerio*), where a total of five *Mcoln* homologs can be found: *mcoln2* and two *mcoln1*- and *mcoln3*-like genes each (Figure 1—phylogenetic tree; navy blue branch) [50].

## 3. TRPMLs—Evolutionary Beginnings

Sequence comparison of the human TRPML1 protein reveals high similarity to TRPML family members in insects [49]. *Drosophila melanogaster* encodes a single TRPML family member that shares 44% amino acid identity with human TRPML1 [12,62].

The functional overlap between *Drosophila melanogaster* Trpml and human TRPML1 becomes particularly evident when comparing associated loss-of-function phenotypes. As previously discussed, loss of TRPML1 in humans causes a lysosomal storage disease marked by prominent mental retardation and impaired motor neuron development [42]. Similarly, disruption of *Drosophila melanogaster* Trpml (*trpml^1^* mutant) results in neurodegeneration and motor defects, resulting from impaired clearance of apoptotic cells from the fly brain. Mirroring MLIV, *trpml^1^* flies exhibited impaired autophagy marked by accumulating macromolecules [63]. It was later revealed that *trpml^1^* pupae exhibit impaired amphisome/lysosomal fusion and impaired mTORC1 activity, attributable to decreased fly viability [64]. The decreased mTORC1 activity was further linked to impaired development of the neuromuscular junction: Loss of Trpml function, and its associated decrease in Rag GTPases and mTORC1 activity, resulted in decreased JNK phosphorylation and, resultantly, impaired synapse development [65]. Excitingly, a high-protein diet (with the co-administration of an ALK inhibitor, permitting neuronal amino acid uptake) was found to suppress LSD-associated neuromuscular junction defects, appearing a potential therapeutic strategy for lysosomal storage diseases [65]. Although the overlap between *trpml^1^* and MLIV is exciting, it should be noted that primary mouse *Mcoln1*^−/−^ neurons isolated do not exhibit abnormal mTORC1 activity, underscoring potential differences between species or neuronal subpopulations [66]. It would certainly be intriguing to compare these findings with the human disease.

Insects in most cases express a single TRPML channel; however, *Ixodes scapularis* and *Tetranychus urticae* express two and three TRPML channels, respectively, implying potential functional divergence [12]. The *Caenorhabditis elegans Mcoln*-homolog is termed *cup-5* and appears ubiquitously expressed in lysosomes. The null mutation of *cup-5* causes maternal effect lethality and lamellar structures similar to those observed in human MLIV cells. Transfection of *cup-5* mutant nematodes with human TRPML1 partially rescues their decreased viability, albeit not as well as overexpression of *cup-5* itself [67]. Lophotrochozoans (*Lottia gigantea* and *Capitella teleta*) appear to be the first species to possess all six metazoan TRP channel subfamilies [12]. Primitive animals, such as sponges (*Amphimedon queenslandica*), Placozoa (*Trichoplax adhaerens*), and Cnidaria (*Nematostella vectensis*), contain only a few cell types, but encode at least one *Mcoln* gene, suggesting the *Mcoln* family already started growing around 600 million years ago [12,68,69,70].

Two choanoflagellates (*Monosiga brevicollis* and *Salpingoeca rosetta*), free-living unicellular and colonial flagellate eukaryotes considered the closest extant relatives of animals, express five TRP subfamilies, among them TRPML [53,71,72] (Figure 2). Furthermore, a TRPML ortholog has been found in *Capsaspora owczarzaki*, demonstrating that many metazoan-type TRP channels emerged already in the unicellular common ancestors of Metazoa. In 2012, a study showed that the apusozoan protist *Thecamonas trahens* expresses TRPP and TRPV channels, suggesting these might represent some of the most ancient metazoan-type TRP channels [53]. TRPMLs and TRPPs have high sequence similarity which may explain why Amoebozoan *Dictyostelium discoideum* located in the evolutionary tree close to *Thecamonas trahens* has both of them. Of note, *Dictyostelium* TRPML is, in contrast to mammalian TRPMLs, required for lysosomal Ca^2+^ uptake [73]. Only TRPP and TRPML have been reported for euglenozoans, *Leishmania* (e.g., *Leishmania major*) and *Trypanosoma* and the alveolate protist *Paramecium tetraurelia*, which evolved around 700 million years ago. In another unicellular eukaryotic protist *Lingulodinium polyedra* TRPMs, TPPMLs and TRPPs have been identified [2,12,54]. In conclusion, TRPML, alongside TRPP, TRPV, and TRPM, seems to have been present in a common ancestor of Unikonts and Bikonts.

## 4. TRPs in Fungi and Plants—Barely Present

Lysosomes and vacuoles share many similarities in molecular composition and function, e.g., degradation of cellular substances, ion accumulation and storage, and residency of luminal hydrolases, membrane transporters, the vacuolar proton pump and ion channels [74,75,76]. Fungi possess one vacuolar TRP channel, Trpy1p, which shows sequence similarity to other invertebrate/vertebrate TRPML channels of around 16% and 40%, respectively [77]. However, Trpy1p does phylogenetically not cluster with any of the metazoan TRP channels [53]. Trpy1p emerged in fungi upon diverging from metazoan ancestors (Figure 1), appearing structurally distinct from the other lysosomal Group 2 TRP channels, lacking the extensive S1–S2 luminal domain (Figure 2). *Saccharomyces cerevisiae* Trpy1p, together with green alga *Chlamydomonas reinhardtii* TRP1, represent the only functionally characterized TRP channels in unicellular organisms [2,55]. Trpy1p acts as a mechanosensor, regulating vacuolar osmotic pressure [55,78]. The well-known mammalian TRPML activator PI(3,5)P_2_ is key to adaptation of yeast cells to osmotic stress. Upon osmotic stress, yeasts activate several channels and permeases via a transient, 20-fold increase of PI(3,5)P_2_, resulting in the release of ions and solutes from the lumen of the vacuole. However, there is no proof for direct activation of Trpy1p via PI(3,5)P_2_ [74,77]. Trpy1p is also linked to increased levels of cytosolic Mn^2+^, which triggers its activation and Ca^2+^ efflux in response to oxidative stress [79].

TRPs are found in mammals, worms, insects, and yeast, but appear mysteriously absent from land plants (Figure 1). In green (*Ulva compressa*) and brown (*Ectocarpus siliculosus*) algae, application of mammalian TRP channel agonists and antagonists leads to a positive and negative regulation of cellular Ca^2+^ uptake, respectively [80,81], suggesting existence of TRP channels in 750 million to 1 billion year old phylae [82]. Furthermore, the single-cell alga *Chlamydomonas reinhardtii*, commonly found in soil and fresh water, has a fine-tuned navigation system based on Ca^2+^ conductance [83,84]. In 2007, Merchant et al. published the genome of *Chlamydomonas reinhardtii* [85] and multiple TRP channels have since been identified [2,56]. Arias-Darraz et al. suggested that sequence similarity as well as phylogenetic reconstruction indicate that CrTRP1 could be considered part of a novel family, structurally appearing a homolog to TRPC channels [86]. Interestingly, more than 80% of the putative TRP channels in algae and unicellular organisms appear to originate in TRPP/TRPML clusters, in agreement with TRPP and TRPV channels found in the protist *Thecamonas trahens* [2,53]. The TRPP/TRPML cluster includes CrTRPP2 and CrTRP11, the latter a distant relative of the TRPV family [56]. CrTRPP2 is structurally more similar to mammalian TRP group 2 channels than *Saccharomyces cerevisiae* Trpy1p (Figure 2). Two independent studies report behavioral changes after knocking down TRP channel transcripts in *Chlamydomonas*. Both CrTRPP2 and CrTRP11 are expressed in the flagella, affecting algae mating behavior as well as mechanosensory responses [56,87]. TRP channels in algae exhibit properties of TRP channels expressed by multicellular organisms, such as weak voltage dependence, activation by temperature, regulation by PI(4,5)P_2_ (phosphatidylinositol 4,5-bisphosphate), and pharmacological block by BCTC or gadolinium [56,86,87,88]. Thus, core functional features of metazoan TRP channels appear present in plant ancestors, suggesting that basic TRP characteristics evolved early in the history of eukaryotes [2].

## 5. TPCs in Plants (Calcium Induced Calcium Release Theory)

In contrast to TRPML channels, TPCs (two-pore channels), the other group of non-selective endo-lysosomal cation channels in mammalian genomes, are widely found in terrestrial (e.g., *Arabidopsis thaliana*) and marine plants (e.g., *Klebsormidium nitens*) [89]. All plants harbor at least one TPC gene, already present in the genome of charophytic algae around 793 million years ago [82,90,91]. TPCs belong to the superfamily of voltage-gated ion channels (VIC), and consist of 12 transmembrane domains (S1–S12) subdivided into two shaker-like domains, each of them containing six transmembrane domains (S1–S6) including the voltage-sensing S4 domain and the ion-conducting pore domain between S5 and S6. TPCs probably originate from a gene-duplication event of single-domain NaV channels [91].

TPC1 activity was first shown by Hedrich and Neher in barley mesophyll vacuoles [92]. Upon activation, plant TPC1 provides an ion-conducting pathway for various cations, mainly K^+^ and Na^+^ [93]. Plant TPC1 (or SV, slow vacuolar channel, its original name) is modulated by several factors, underpinning its complex regulation. Beside voltage, TPC1 is regulated by calmodulin [94], 14-3-3 proteins [95], kinases and phosphatases [96,97], pH [93,98], redox state [99], and Mg^2+^ and Ca^2+^ [92,100]. In addition, natural polyamines (e.g., spermidine [101,102]) and heavy metals [103] have been reported to inhibit ion passage through open TPC1 channels in red beet and radish.

Since loss of TPC1 function does not drastically impair plant growth [104], its physiological role is a matter of debate. However, roots of seedlings exposed to salt treatment show reduced growth in the TPC1 knockout *tpc1-2* mutant compared to WT plants [105]. In contrast, TPC1 overexpression increases salt tolerance [105]. Interestingly, salt-triggered propagating Ca^2+^ signals in the root were attenuated in *tpc1-2* mutants, but increased in TPC1 overexpression lines [105]. Furthermore, it was shown that systemic Ca^2+^ signals, generated upon wounding, were gone upon loss of TPC1 function. This observation pointed to a role of TPC1 in systemic Ca^2+^ signaling [106]. The *Arabidopsis thaliana* TPC1 *fou2* variant (fatty acid oxygenation upregulated 2) point mutation D454N leads to an increased production of the stress hormone jasmonate, even under non-stressed conditions. The *fou2* plants exhibit a strong growth retardation phenotype [107,108], probably originating from the increased vacuolar K^+^ release due to TPC1 hyperactivity [108] It is important to note that a TPC1-independent pathway of jasmonate signaling has also been postulated [109]. Since TPC1 participates likely indirectly in the generation/modulation of the Ca^2+^ wave, it seems to be reasonable to suggest a supreme trigger, regulating Ca^2+^ and K^+^ fluxes [109]. Vacuolar membrane depolarization may be one of the missing early triggers for jasmonate production. Furthermore, TPC1 is a prerequisite for vacuole membrane excitability [110], thus triggering of vacuolar membrane depolarization in local spots may be an elaborate way to encode more complicated information in long- and short-distance signaling pathways in plants.

The concept of Ca^2+^-induced Ca^2+^ release (CICR, calcium induced calcium release), initially proposed by Fabiato et al. (1985) in the animal field, was adapted by Ward and Schroeder (1994) to plant research [111,112,113]. Based on patch-clamp measurements, they postulated cytosolic Ca^2+^ signals generated by TPC1 in *Vicia faba* guard cell vacuoles. However, the ionic condition used in this study was far away from the physiological concentration for Ca^2+^ and K^+^. By applying non-physiological ionic conditions, TPC1 channel-mediated Ca^2+^ currents were also recorded in other species, but only in the opposite direction, from cytosol to vacuole [114,115,116]. Furthermore, an inhibitory effect of vacuolar Ca^2+^ was postulated [117], likely attributable to the highly conserved vacuolar Ca^2+^-binding motifs of TPC1 [91,118]. To solve the above long-lasting debate, structural models for the different species will be helpful. Of note, the gain-of-function *Arabidopsis thaliana* TPC1 channel variant (*fou2*) shows increased vacuolar Ca^2+^ and slightly lower resting cytosolic Ca^2+^ levels compared to WT, which would be difficult to reconcile with TPC1 releasing Ca^2+^ under physiological conditions [98,109]. In sum, the contribution of plant TPC1 to global as well as local Ca^2+^ signals remains debated and needs to be further evaluated. A similar complex debate exists in the field of mammalian TPC research, where the role of TPCs in endo-lysosomal Ca^2+^ release remains controversially discussed.

## 6. TPCs in Metazoa

The general structure of metazoan TPCs is very similar to their green orthologs. The unique TPC1 gene structure (2 × 6 TMDs) contrasts the 1 × 6 TMD channels (e.g., voltage-gated potassium channels or TRP channels) and the 4 × 6 TMD voltage-gated calcium or sodium channels [119,120]. This TPC-like architecture has been found also in unicellular eukaryotes such as diatoms or amoebae, but not in prokaryotes [91]. Remarkably, TPC genes in fungi seem to be lost immediately after separation from the animal lineage (Figure 1) [53]. However, a variable amount of TPC orthologs was found in choanoflagellates (e.g., *Monosiga brevicollis* [57]) and even in the evolutionarily older Filaserea and Apusozoa—*Capsaspora owczarzaki* and *Thecamonas trahens*, respectively [53]. The oldest living animals, sponges, have six homologs of the TPC gene, a higher number compared to other Metazoa (Figure 1). Despite the absence of the TPC1 gene in *Drosophila melanogaster* and *Caenorhabditis elegans*, other Protostomes such as Lophotrochozoans (*Lottia gigantea* and *Capitella teleta*) encode three orthologs of human TPCs. Deuterostomes such as sea urchin (*Strongylocentrotus purpuratus*) also possess three TPC isoforms (Figure 1; [121]). Intriguingly, only TPC1 and TPC2 channels are functional in humans and rodents, while TPC3 is a pseudogene in humans and some primates, and it is completely missing in rodents [25]. Two-pore channels 1–3 are localized in endo-lysosomes, but *Danio rerio* TPC3 can also function in the plasma membrane [122,123]. TPCs form a subfamily of eukaryotic voltage- and ligand-gated cation channels, however the proposed main ligands in plants and animals differ: calcium in plants, compared to phosphatidylinositol (3,5)-bisphosphate (PI(3,5)P_2_) and nicotinic acid adenine dinucleotide phosphate (NAADP) in animals [92,124,125]. Originally, TPCs were proposed as ion channels involved in NAADP-mediated Ca^2+^ release from intracellular acidic stores in animals [124]. It has been suggested that TPC-induced Ca^2+^ release from acidic stores can induce downstream Ca^2+^ release from the ER through Ca^2+^ induced Ca^2+^ release (CICR) [126]. It has also been postulated that mammalian TPCs are not Ca^2+^ release channels activated by NAADP, but rather Na^+^ release channels activated by PI(3,5)P_2_ [23,31]. The discovery of novel lipophilic small molecule agonists for TPCs in two independent high-throughput calcium imaging campaigns [127,128] partially reconciled this debate. Both campaigns revealed that TPC agonists can be identified in calcium imaging experiments, suggesting that TPCs are indeed permeable to Ca^2+^. Furthermore, agonists were found to differentially affect the Ca^2+^/Na^+^ permeability ratio, suggesting that the cation permeability of the channel can be modulated in a ligand-dependent manner, in turn resulting in distinct effects on key lysosomal functions such as lysosomal pH and exocytosis [127].

Loss of TPCs seems to not affect viability, as double TPC1/2 KO mice age normally [129]. There are also no disease-related (point) mutations known for human or rodent TPCs [31]. However, channel function becomes essential under stress conditions such as decreased availability of ATP and nutrients or other challenging conditions such as infection with viruses or bacterial toxins, or certain diets (e.g., high cholesterol diet) [23,25,130,131].

Evidence for a role of TPCs in Ca^2+^ signaling is also found in other species. Kelu et al. (2018) demonstrated that in zebrafish Ca^2+^ release via TPC2 from acidic stores/endo-lysosomes is required for the establishment of synchronized activity in primary motor neurons (PMNs), and postulated a role of NAADP/TPC/Ca^2+^ signaling in skeletal muscle differentiation [132]. Furthermore, Andrew Miller’s group discovered a putative link between ARC1-like (zebra fish ADP ribosyl cyclase (ARC) homolog) and NAADP generation, TPC2, and Ca^2+^ signaling during zebrafish myogenesis. Knock down or pharmacological inhibition of ARC1-like leads to an attenuation of Ca^2+^ signaling and disruption of slow muscle cell development [133]. In single- or multi-celled phagotrophic bacterivores, *Dictyostelium discoideum* TPC2 disruption leads to delayed development and prolonged growth in culture, delaying expression of early developmental genes. Extracellular cAMP-induced Ca^2+^ signals are delayed in *tpc2*-disrupted cells, and sensitivity to weak bases is increased, consistent with an increase in vesicular pH [134].

Despite extensive support for the physiological relevance of TPCs in endo-lysosomal Ca^2+^ signaling, several questions remain surrounding TPC function. It remains to be further elucidated how NAADP activates TPCs, directly or indirectly, and what the physiological consequences of ligand-dependent cation permeability changes in mammalian TPCs are. Furthermore, the physiological role of Na^+^ release from mammalian TPCs remains unclear, and the up- and downstream signaling pathways of TPCs need to be further elucidated.

## 7. Summary

Both TRPMLs and TPCs are comparatively old proteins. They coexist in many species with some remarkable exceptions, e.g., TPCs are missing from *Caenorhabditis elegans* and *Drosophila melanogaster*, while they are in contrast to TRPMLs present in most plants. Nevertheless, coexistence is highly conserved throughout the Metazoa kingdom and beyond. Unlike mammalian TPCs, plant TPC1 is a Ca^2+^-regulated, nonselective cation channel, insensitive to PI(3,5)P_2_ and NAADP [135]. The reasons for such divergent evolutionary developments are not known. In plants, this may be correlated with the complete loss of TRP channels already at early stages of plant evolution. While during evolution, the number of TPC isoforms has steadily decreased from seven in *Thecamonas trahens* and six in *Amphimedon queenslandica* to zero in insects and only two in humans, the number of Mcoln genes seems to increase, reaching three and even five isoforms late in evolution (*Callorhinchus milii* and *Danio rerio*), suggestive of a possible increasing physiological relevance of TRPML channels during evolution.

## Figures and Tables

**Figure 1 ijms-21-04181-f001:**
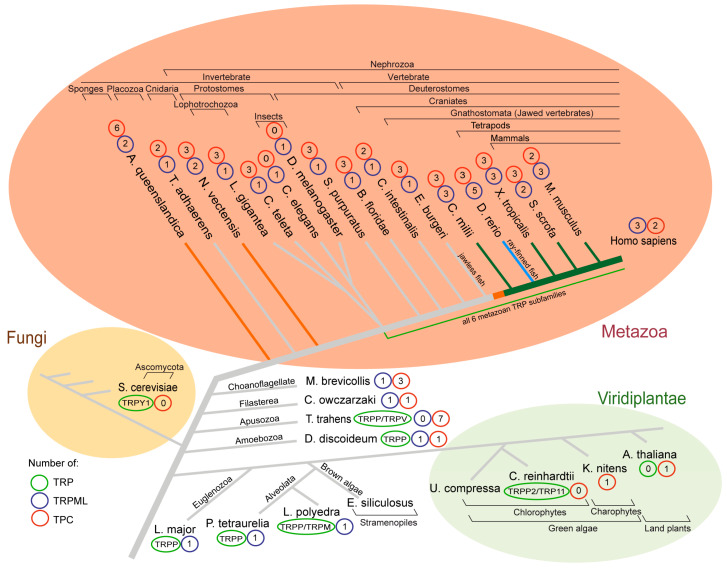
Phylogenetic tree indicating the number of genes encoding TRPML and TPC proteins. Numbers of TRP, TRPML, and TPC proteins of selected species of the Metazoa, Fungi, and Viridiplantae as well as selected ancestral species. The tree was constructed based on information included in the following resources [2,12,48,50,51,52,53,54,55,56,57,58,59,60,61], and constructed manually based on this information. Branch color coding of the tree indicates a first (orange), second (bright green), and third (blue) gene duplication of TPRML. Full species names can be found in the text.

**Figure 2 ijms-21-04181-f002:**
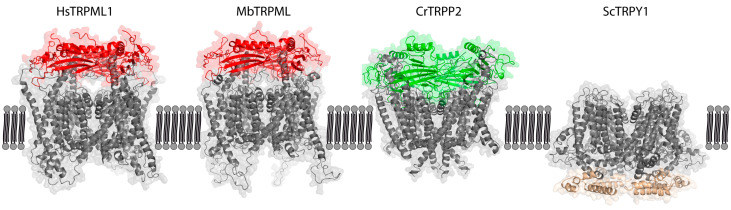
Main structural differences in loops linking TMD between TRPML ancestors of animal (red), plant (green), and fungi TRPY1 (brown) kingdoms. Amino acid sequences were obtained for *Monosiga brevicollis* MbTRPML (A9UQ01), *Chlamydomonas reinhardtii* CrTRPP2 (ABR14113), and *Saccharomyces cerevisiae* ScTRPY1 (Q12324), and aligned to resolved proteins using SWISS-MODEL (https://swissmodel.expasy.org/). Homology models were constructed using 6BCO (*Mus musculus* TRPM4) as a template for ScTRPY1, 6T9N (*Homo sapiens* Polycystin-2) for CrTRPP2 and 6E7P (*Homo sapiens* TRPML1) for MbTRPML, and visualized using PyMOL. Only MbTRPML aligned well with human TRPML1. Red and green structures indicate the mucolipin or polycystin domains of TRPML and TRPP, respectively, while the TRPY1-unique intracellular domain of ScTRPY1 is colored brown. Having constructed the homology models, we interrogated the possibility of vacuolar channel sorting. Putative lysosomal targeting signals (LTS) were identified using a custom Prosite scan (https://prosite.expasy.org/cgi-bin/prosite/), searching for motifs DxxLL, [DE] xxxL [LI], DxxLM, DxxMV, IMxxYxxL (plant), and YxxL (yeast). Only cytosolic LTS were considered relevant. TRPY1 bears three yeast LTS (Y107-L110, Y293-L296, and Y473-L476, YxxL), while CrTRPP2 bears two yeast and one cross-species LTS (Y44-L47 and Y95-L98, YxxL; E1459-L1464, [DE] xxxL [LI]). MbTRPML bears a single cross-species LTS (E500-I505, [DE] xxxL [LI]).

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
