# Peer review of "Evolutionary Aspects of TRPMLs and TPCs"

_ijms, 2020, doi:10.3390/ijms21114181_

Round 1
Reviewer 1 Report
JaÅ›lan et al review and discuss TRPML and TPC channels in diverse species. The authors have covered many TRPML and TPC homologs that exist from single-cell organism all the way to plants and humans. My main concerns are the resolution and source of the two figures: 1) The resolution is too low to read the texts on the figure; 2) Are the figures produced originally for the manuscript? There is a “(a)” shown at the bottom of Figure 1, leading me to suspect that it was captured from other previously published source. Please also stated in the figure legends what software/program was used to make the diagrams. In addition, for readers not in the TRP field, the nomenclatures of Mcoln and TRPML might be confusing, especially when “Mcoln” appears in the text while “TRPML” in the figure. I suggest sticking with TRPML for all organisms except for humans, where MCOLN1 gene was named because it causes the disease.
Other specific comments are as follows:
- Line 74: What does it mean by “counteracts v-ATPase activity”? Cation efflux and anion influx have been recognized as facilitating lysosome acidification. Also, when referring to lysosomal membrane potential, the directionality should be defined, e.g. cytosolic side being zero mV. The authors should also mention Wang et al. “A voltage-dependent K+ channel in the lysosome is required for refilling lysosomal Ca2+ stores” (JCB 2017) which reported a positive lysosomal membrane potential as opposed to the negative values reported by others.
- Line 110: “…localization of Mcoln2 of (and) Mcoln3 genes…”
- Line 116: I don’t see any phylogenetic branch in navy blue color. Please check with the production editor to make sure the colors are accurately reproduced. Also, the citation of Ref#48 is not for Figure 1 and should be placed outside of the brackets.
- Line 128-130: When reviewing about TRPML function in genetic model organism, the recapitulations and resemblances of pathological phenotypes in Drosophila model should be mentioned. For instance, Venkatachalam et al (Cell 2008), Wong et al (Curr Bio 2012), and Wong et al (Cell Rep 2015) have reported cellular and physiological phenotypes that closely resemble those in humans with MLIV.
- Line 134: Which “human Mcoln gene” rescued the worm phenotypes?
- Figure 2: Only the 3D reconstruction of the 4 TRPML species are shown in Figure 2. Where are the homology models? What does the color coding represent? Where is the diagram showing the dileucine and tyrosine motifs in different TRPML species?
- Line 211: What does “(CICR- theory)” mean? Are plant TPCs a hypothetical CICR channel?
- It would be very informative to have a table categorizing the diverse TRPML and TPC in the organisms that are discussed in the manuscript. Features like number of paralogs, ion selectivity, and cellular/biological functions can be listed, along with reference citations.
- Line 313-317: Please correct the grammar and structure of this one sentence-long paragraph.
Reviewer 2 Report
This is a brief review focusing on the evolutionary aspects of TRPML channels and TPC channels, it summarizes the phylogenetic of the encoding genes. Being a pharmacologist I would like to see much more pharmacology and pathophysiology about these channels, but I understand that this is not the aim of this review.
I suggest the authors however to briefly discuss some more pathophysiology and introduce some references discussing these aspects, for instance
Cancers (Basel). 2019 Jan; 11(1): 27. Published online 2018 Dec 27. doi: 10.3390/cancers11010027 PMCID: PMC6356888 PMID: 30591696 or others in the literature
Round 2
Reviewer 1 Report
The authors have adequately addressed my concerns.